# Current Challenges Supporting School-Aged Children with Vision Problems: A Rapid Review

**Qasim Ali** [1] , **Ilona Heldal** [1,*] , **Carsten G. Helgesen** [1] , **Gunta Krumina** [2] , **Cristina Costescu** [3] , **Attila Kovari** [4] , **Jozsef Katona** [5] and **Serge Thill** [6]

1   Department of Computing, Mathematics, and Physics, Western Norway University of Applied Sciences, Inndalsveien 28, 5063 Bergen, Norway; ali.qasim@hvl.no (Q.A.); carsten.gunnar.helgesen@hvl.no (C.G.H.)
2   Department of Optometry and Vision Science, University of Latvia, LV-1586 Riga, Latvia; gunta.krumina@lu.lv
3   Special Education Department, Faculty of Psychology and Educational Sciences, Babes-Bolyai University, Sindicatelor Street No 7, 400029 Cluj-Napoca, Romania; christina.costescu@gmail.com
4   Department of Natural Sciences and Environmental Protection, Institute of Engineering, University of Dunaujvaros, Tancsics M 1/A, 2400 Dunaujvaros, Hungary; kovari@uniduna.hu
5   Department of Software Development and Application, Informatics Institute, University of Dunaujvaros, Tancsics M 1/A, 2400 Dunaujvaros, Hungary; katonaj@uniduna.hu
6   Donders Institute for Brain, Cognition, and Behaviour, Radboud University, Thomas van Aquinostraat 4, 6525 GD Nijmegen, The Netherlands; serge.thill@donders.ru.nl
*   Correspondence: ilona.heldal@hvl.no; Tel.: +47-5558-7524

**Abstract:** Many children have undetected vision problems or insufficient visual information processing that may be a factor in lower academic outcomes. The aim of this paper is to contribute to a better understanding of the importance of vision screening for school-aged children, and to investigate the possibilities of how eye-tracking (ET) technologies can support this. While there are indications that these technologies can support vision screening, a broad understanding of how to apply them and by whom, and if it is possible to utilize them at schools, is lacking. We review interdisciplinary research on performing vision investigations, and discuss current challenges for technology support. The focus is on exploring the possibilities of ET technologies to better support screening and handling of vision disorders, especially by non-vision experts. The data orginate from a literature survey of peer-reviewed journals and conference articles complemented by secondary sources, following a rapid review methodology. We highlight current trends in supportive technologies for vision screening, and identify the involved stakeholders and the research studies that discuss how to develop more supportive ET technologies for vision screening and training by non-experts.

**Keywords:** school children; functional vision; vision screening; vision training; eye-tracking; stakeholders





## 1. Introduction

Our vision system is essential for performing daily activities and interactions with the environment. The vision problems that can affect a child's vision are divided into two groups: eye disorders in which the eye does not focus the light that enters, resulting in blurred vision, such as myopia, amblyopia, strabismus, and eye diseases that are caused by changes in eye structure, such as a cataract or retinoblastoma. Uncorrected refractive error is a significant cause of blindness, and the major cause of impaired vision in both industrialized and developing countries [1,2]. Indeed, 99.2 million children around the world under 15 years of age have visual impairment (visual acuity < 6/10) [3]. A vision problem can occur even if eye health seems normal or does not show any signs of binocular disorders [4]. There are many children, not only with functional vision problems or eye diseases, but also with dyslexia and other cognitive impairments often associated with vision problems [5,6]. The symptoms of these problems can be similar, and there are children with multiple problems. Scheiman associates such problems with visual

information processing due to physical or cognitive deficiencies [7]. ADHD and dyslexia are examples of cognitive disorders that are often associated with vision problems; they have the same symptoms, but need to be treated separately [5,8,9]. The early detection of vision disorders is also important for minimizing upcoming problems associated with academic performance and social life [10–12].

Vision screening is compulsory in many European countries, and is generally performed on children prior to the commencement of school education at the age of 4–5 years [13]. After this age, the parent or legally responsible person has to follow up on their child's vision health. Since children's vision continues to develop, it may change and they may experience difficulties, especially when they begin at school [6,14,15]. If teachers or parents observe problems, the most important stakeholders for screening and helping with treatments are the responsible vision specialists from healthcare institutions. Even visual functional assessments (e.g., visual acuity, color vision testing, and contrast sensitivity) can fail to diagnose the possible functional problems of the vision system [16]. These not only relate to reading difficulties, but have several symptoms, such as deficient attention span, fatigue, headache, dizziness, poor academic performance, and productivity [17–19]. A child with some of these problems can often control eye movements for a few minutes before experiencing difficulties. Therefore, during typical tasks with vision screenings—which often have a short duration—these problems may not be identified [20]. Some of these problems can be due to various concrete functional problems of the vision system, and are alleviated through vision training [21,22]. A possibility to help children with their vision would be allowing screening for all when they begin at school. Since the number of vision experts performing the screening is low, for this, it would be necessary to involve non-experts.

In recent years, promising options supporting necessary vision tests have begun to appear that use technologies and computerized tasks [20]. Several eye-tracking (ET) technologies can measure the movements of the left and right eye separately, and therefore ET technologies are promising for producing a measurement that can help to indicate functional vision problems [23]. ET technologies are often integrated with computer programs to make screening and training more flexible by incorporating serious games (SG), or utilizing virtual reality (VR) technologies to use the surrounding space [24]. However, it is difficult to keep track of and understand the role of these new solutions.

The motivation behind this paper is to enhance the understanding of what technology development can contribute to screening the vision of school-aged children, especially for non-vision expert users. For this, one needs to know who can be involved in vision screening, their roles, competencies, and willingness. Their work needs to complement or be aligned with professional vision experts' work, and possibly be assisted with tools and technologies providing evidence for further actions if needed. For developing new supporting technologies, it is not enough to understand who can use it (i.e., the possible non-vision experts) and for whom (i.e., the school-aged children). The knowledge of vision experts is necessary to determine how screening by non-experts can be quality assured, and how screening by non-experts needs to be communicated with vision experts.

The overall aim of this paper is to identify current trends in the research that is needed to enhance technological support for vision screening and training and involve non-vision expert stakeholders. This will be completed via a rapid review [25] based on an iterative approach to mapping evidence with knowledge gaps from highly multidisciplinary domains, such as those practicing vision science, special education, cognitive science, and technology management and development. While it is difficult to perform a full and systematic review at this stage, some basic ideas originating from mapping studies [26], used for answering the questions defined in Table 1, also influenced this research.

**Table 1.** Questions defined for the rapid review.

| ID | Research Question | Rationale (Is To) |
|---|---|---|
| RQ1 | Who are the stakeholders influencing vision screening at schools, and what are their roles? | Identifying the involved vision experts and non-experts influencing vision screening in children. |
| RQ2 | What is known about children's vision screening in schools? | Exploring current methodologies and procedures that exist for children in school. |
| RQ3 | How can ET better support screening and training vision? | Investigating the evidence of eye-tracking technologies to support stakeholders in vision screening and training. |

This paper reviews literature about screening and helping school-aged, 6–15 year old children. The main focus is not on vision experts from healthcare (e.g., ophthalmologists or optometrists) or special pedagogy, but on other stakeholders without vision competence who can help to identify vision problems in children (e.g., school nurses, curators, teachers). While non-expert stakeholders can seldom make a diagnosis, they can produce clear indicators about possible issues, and communicate these to a child's parents or legally responsible person. As the next step, these indicators will support communication with healthcare institutions. Indeed, technologies and computer programs can further help vision experts, but the primary focus is not on technologies supporting vision experts.

While ET technologies have existed since the last century [27], affordable ET technologies are are available only during the last two decades, and therefore we do not limit the search to a specific period. To provide a context for better potential vision care in school, we aimed to position functional vision disorders on the map of broader vision problems and at schools [28].

## 2. Methodology

Identifying issues from three different domains (health, education, and technology development) and informing about the need for further state-of-the-art technology development for screening all school-aged children at schools, involving even non-vision experts, would require policy changes. This remains the next step of the research. The intention here is to define the necessary preconditions, such as who influences the handling of vision problems, and how technologies can better support their work.

To map all involved stakeholders and technologies, we performed this rapid review, following the eight phases defined by Khangura et al. [25] and the process described in Figure 1. The main reason for this approach was the need to bring together these broad domains and identify essential gaps that needed to be overcome during a limited timeframe. This approach considered the strengths and limitations defined by Gant [29]. One of the main strengths is the possibility of considering these three broad domains in the review. The primary limitations concern possible missed studies and difficulty contrasting them, which this paper has balanced with a strict documentation of the process.

While the timeframe for a rapid review is much shorter than for systematic reviews, the process is still based on clear questions, chosen criteria, and is rigorous. Instead of quantitative results, it ends with a descriptive summary of categories important for the research questions. This paper utilizes the rapid review methodology suggested by Khangura et al. [25], extracts and summarizes evidence from different domains, and brings knowledge to practice in eight steps. The eight phases are defined as follows: (1) needs assessment, (2) question development and refinement, (3) proposal development and approval, (4) systematic literature search, (5) screening and selection of studies, (6) narrative synthesis of included studies (including evidence), (7) report production, and (8) ongoing follow-up and dialogue with knowledge representatives from the involved domains (see Figure 1).

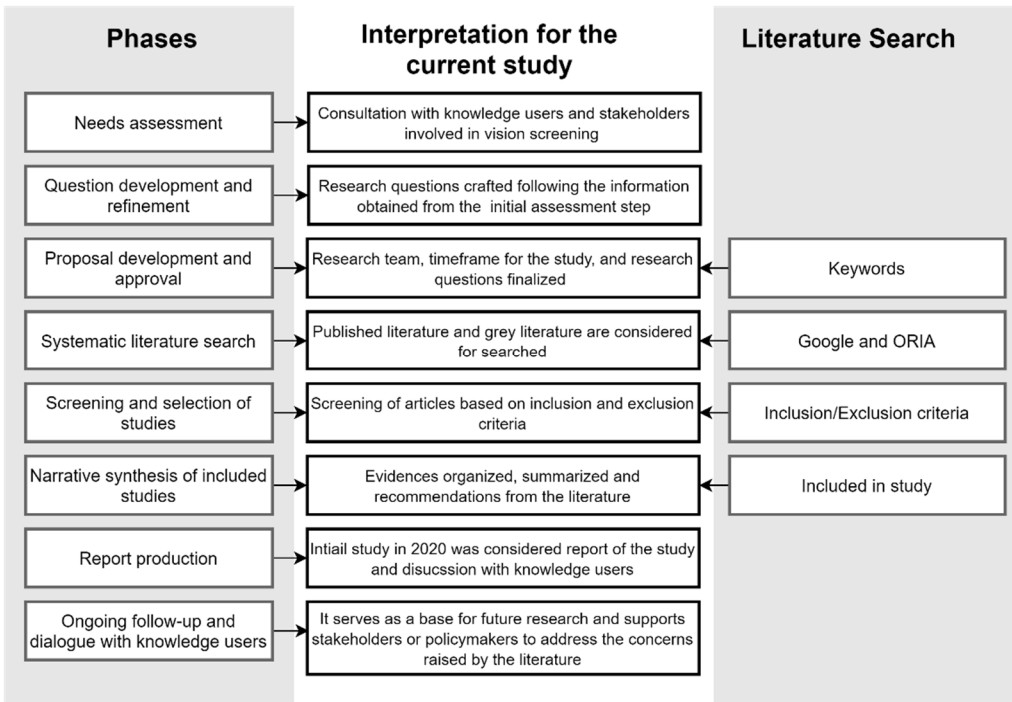

**Figure 1.** Eight steps iteratively considered for rapid review and their relationship within the context of current study and literature search.

The first two phases were completed in the last five months of 2020 [30], resulting in the three research questions that are synthesized in Table 2. The second phase for proposal development was also influenced by how mapping studies investigates literature and grey literature to map already existing research to research questions [31]. Phase (3) is realized through discussions with clinician vision experts, experts in special pedagogy, cognitive- and neuro-psychology for school beginners, and technology developers; this defines the search strategy and primary sources for research works (see Section 2.1). Phase (4) identifies and delimits the research studies, and presents the narrative in the flowchart in Figure 1 (for presenting phases (5) and (6), see Section 2.2). Phase (7) resulted in an earlier initial report from this study [30], which was followed by a discussion with knowledge users (phase (8)). For this rapid review, we use the term "phase" (not steps, as the original research by Khangura et al. [25] calls these) to distinguish from the major steps guiding the literature search (last column in Figure 1).

**Table 2.** Keywords guided the search of the literature for each research question.

| Research Questions | Main Keywords |
| --- | --- |
| RQ1 | "Vision screening" AND "school" AND "stakeholders" |
| RQ2 | ("vision screening" OR "vision assessment") AND "school children" AND "oculomotor dysfunction" |
| RQ3 | ("vision screening" OR "vision assessment") AND "oculomotor dysfunction" AND "eye-tracking" |

### 2.1. Search Strategy

Google Scholar and HVL Library (ORIA) [32] search engines were used to search the articles from domains answering the research questions. The HVL Library has access to more than 160 resources and databases, including MEDLINE, BMC, ACM, IEEE, Web of Science, and SCOPUS. We identified literature on vision screening from 15 multidisciplinary sources of clinical, engineering, and education journals and conferences, including "*Neural Syst Rehabil Eng*", "*Teaching Exceptional Children*", "*E-Health and Bioengi-neering Conference*"

*(EHB)"*, *"Virtual Rehabilitation (ICVR)"*, *"Child: care, health, and development"*, *"American Orthoptic Journal"*, *"Pediatrics"*, *"BMC Medical Education"*, *"Software Engineering (JCSSE)"*, *"Australian Journal of Education"*, *"Survey of Oph-thalmology"*, *"Conference on Cognitive Info-communications"*, *"The Future of Educa-tion"*, *"Journal of Behavioral Optometry"*, and *"Serious Games and Applications for Health (SeGAH)"*.

The keywords used in the search process are defined in Table 2. After the search, we checked the title, keywords, and abstract to examine the cited papers of relevant articles. The search strategy also includes the selected papers' reference lists. If we found an author with important papers, such as [33], we checked related publications from their environment. Examples of such manual selection included recently published articles involving practitioners [34,35], while the EU database for current vision screening projects [11,13] was further explored for grey literature. Since the authors are experts in different domains, they suggested literature or checked others for representativeness of their own domains.

*2.2. Inclusion and Exclusion Criteria, the Narrative and Study Selection*

The following were the inclusion criteria for selected articles:

(1)  Study included vision screening.
(2)  Stakeholders involved in vision screening.
(3)  Oculomotor dysfunction assessment.
(4)  Written in the English language.

Exclusion criteria were as follows:

(1)  Studies intended for hearing, cognitive, or related to any other disabilities.
(2)  Severe vision problems such as blindness and retinoblastoma.
(3)  Use of eye-tracking technologies for purposes other than vision screening or rehabilitation.

The results included a large number of articles, including review papers, mainly from the fields of medicine, education, applied sciences, optometry, and ophthalmology. The initial number of papers was 1049 from Google-Scholar, and 185 from O-HVL. The number of examined papers that we decided to further examine was 192, including eight review papers and four survey papers based on analyzing their title and reviewing the abstract by two authors. After the initial selection, 142 articles were considered relevant for this review. The descriptive results begin with Sections 3 and 4, positioning the problem area while bridging the domains that handle practical vision screening and training to technology development, and connecting to the literature findings and answering the three research questions in Section 5 (RQ1), Section 6 (RQ2), and Section 7 (RQ3).

## 3. Overall Context of Vision Screening

As we described earlier, there are vision problems influencing the quality of life and development of individuals. Presently, vision problems are detected by traditional and instrumental screening, often by professionals. Most detected vision problems can be resolved by corrective lenses (spectacles or contact lenses), vision training, or eye surgery [36]. Children's vision is essential when starting school; 70% to 80% of all tasks in an educational setting require good vision, yet children's vision is seldom screened at this stage [37,38]. Many children may be disadvantaged by not enjoying the same learning conditions as children with no vision problems, or those having previously recognized vision problems. The level of income, parental education, the policies of countries, and ethnicities are identified as factors influencing whether school-aged children benefit from vision screening or not [28,35,39].

Vision screening is usually carried out using a visual attention task that requires children to judge how much of the visual world they see in clinical and non-clinical settings. During the task, the children's visual attention may drop, and they may not be able to maintain their focus [40]. If the children cannot keep their attention, they may stop looking at the given task and become bored. Consequently, their responses become less accurate and sometimes they can be misdiagnosed [41]. Occasionally, attention may drop

due to other problems, such as ADHD or dyslexia [42], problems associated with vision difficulties [43–45]. Vision problems are often associated with hearing or other motoric problems, such as balance [24,46].

Generally, vision screening tests are carried out in primary care clinics by practitioners for checking eye health in children. Some of the tests can be supplemented by technologies. Metsing et al. [47] identified three categories for vision screening, namely, conventional (traditional or manual), instrumental, and computer software (see Figure 2). Her categories can be extended for vision training as well, and the second and the third categories can also overlap each other. In this paper, an additional category is differentiated (Section 3.4.) for web and smartphone-based software tools; the discrimination of this category often overlaps with other computer-based programs. However, this category points to newer, more affordable, and portable possibilities that appear today.

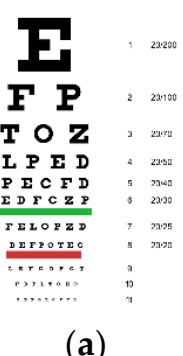

(a)

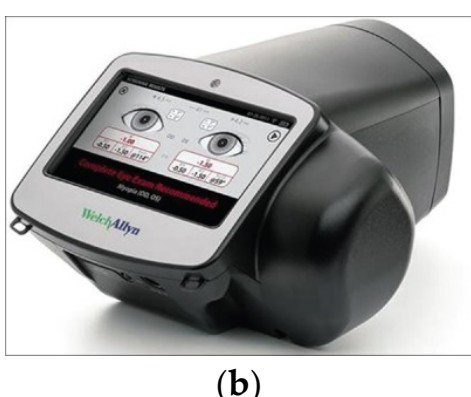

(b)

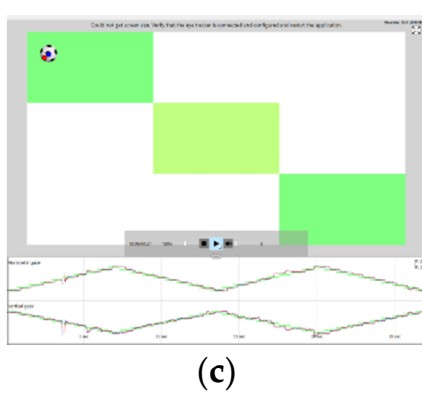

(c)

**Figure 2.** Different tools and computer software programs supporting different phases in a vision screening: (**a**) Snellen chart used for traditional vision screening (image downloaded from https://en.wikipedia.org/wiki/Snellen_chart#/media/File: Snellen_chart.svg, accessed date 12 October 2021); (**b**) Welsch Allyn Spot Vision Screener is an instrumental device used for measuring refractive error (image downloaded from https://www.gocheckkids.com/welch-allyn-spot-vision-screener, accessed date 12 October 2021); (**c**) C&Look is an eye-tracking computer software for screening oculomotor problems.

*3.1. Conventional (Traditional, Manual) Screening*

The traditional way of diagnosing vision problems among children and adults is by "manual" vision screening during which vision specialists examine vision problems through several tests. Specialists define the procedure (the included tests and how the chosen tests are combined in a test battery). Already, these procedures use several aids, e.g., the Snellen chart (see Figure 2a), the plus lens test, and stereo test [48]. These are aids helping to investigate problems such as reduced visual acuity, hypermetropia, and binocular vision. Traditional methods of vision screening are considered gold standard procedures. As such, they have the advantage of having been tested and approved extensively by now. However, administering them requires expert knowledge from vision specialists, or specially trained school nurses, or educated volunteers.

In most countries, systematic screening is implemented around the age of four or five [13] as a last recommended and state-financed vision screening. This screening seldom includes functional vision screening, e.g., screening for identifying ocular motility disorders or other problems with the vision system in one's life [6,49]. The common vision problems occur in preschool- and primary school-aged children, and the main focus of most screening programmes is to detect amblyopia, refractive errors, binocular vision, and ocular motility disorders (e.g., strabismus) [47].

Meanwhile, the most common vision problems in primary school- and secondary school-aged children are uncorrected refractive error [50], oculomotor dysfunction (an anomaly in fixation, saccades, or pursuit eye movements) [51], and accommodative and non-strabismic binocular disorders (e.g., accommodative insufficiency, convergence insuffi-

ciency) [52,53], especially in learning disabilities and children with ADHD [54]. However, visual screenings to assess this visual functionality do not exist.

### 3.2. Instrumental Screening

Instrumental screening methods are already appropriate for children 1 or 2 years old, and are recommended for children older than 2 years old under certain conditions [55]. Instrumental vision screening includes supported instruments from the available automatic vision screening devices that use state-of-the-art technologies, e.g., autorefraction, retinal birefringence, or a photo screener [56]. Automated vision screening devices are used primarily for diagnosing refractive errors, media opacities, and misalignments of eyes in children and adults [57] (see Figure 2b).

Instrumental technologies show high accuracy in identifying vision problems in children. Silverstein et al. [58] showed that photoscreener-based instrumental technology could detect refractive errors in children (particularly useful in younger children (ages 3–5), preverbal children (under age 3), and nonverbal children) that may be missed in traditional screening methods. The speed of vision screening is almost three times faster than the traditional method. Vaughan et al. [59] compared the results obtained via photoscreening technology with traditional chart methodology to identify amblyopia and reflective errors in preschool children. The results showed 82.9% of the children passed instrumental eye exams, while this rate was 64% through LEA chart-based methodology [59].

The acquisition of useful visual field measurements in children, particularly younger children, is difficult and sometimes impossible. Children with moderate and severe visual impairment, glaucoma, damaged brain, or a central nervous system abnormality, such as perinatal stroke or optic pathway glioma, should undergo visual field testing. The biggest challenge in visual field testing is recognizing when a child may have a visual field defect, especially if it combines with ADHD. Vision specialists and researchers commonly use standard automated perimetry (SAP) in clinical settings for visual field testing [60]. The accuracy of visual field testing can be influenced by several factors, including time to respond, attention, fatigue, and stimulus size and duration [61]. Akar et al. [62] investigated the visual field testing strategy for healthy school-aged children with the Goldmann perimeter and Humphrey Field Analyzer. In order to avoid fixation loss, different rewards, verbal instruction, and gamified response buttons were given to children. The results showed that children faced difficulties with Goldmann kinetic perimetry tests. They concluded that the visual field of children could be examined using static perimetry at the age of six. However, visual field assessment is harder to perform in children with AHDH due to the lack of comprehension of the testing method, loss of gaze on the fixation target, rapid boredom, fatigue, and distraction [63]. In this case, eye-tracking technology is most suitable to compensate for inadequate eye fixation control during perimetry [64].

The use of photo screeners allows for the quick, non-invasive measurement of refraction and ocular alignment in both eyes, and would be of great value in refractive error screening, early detection of amblyopia, and in eye care practice and research [65]. Unfortunately, there is no possibility to assess the visual function (visual acuity, color vision, stereovision etc.)

### 3.3. Computerized Vision Screening Programs

Computerized vision screening programs offer a broad range of test batteries for vision screening, including visual acuity, color vision, contrast vision, stereovision, visual efficiency skills, visual field, test, and oculomotor behavior of the eyes [66,67]. For example, Visual Efficiency Rating (VERA) is a computerized software developed for school nurses to screen visual acuity, saccades, and the accommodative and vergence facility in children [66]. VERA simulates reading by placing numbers or text sequentially in a pattern while performing the vision test on children [68]. However, the disadvantage of VERA is the low range of sensitivity, which could miss children with vision problems. Gallaway

et al. [69] observed that the sensitivity of VERA ranges from 45% to 64% on school children with visual acuity of 20/25 or better.

In recent years, there has been an increasing interest in using eye-tracking computer programs to screen oculomotor behaviors of the eyes by reading or looking at the animated objects on a computer screen. RightEye [35] and C&Look [23] are the latest developments in the quest to create a new generation of software that can detect the signs and patterns of oculomotor problems (see Figure 2c).

As mentioned earlier, the accuracy of visual field testing can be lowered due to several factors, especially alertness and the inappropriate responses of children in the perimeter. Miranda et al. [70] developed a computerized solution for pediatric visual field tests using computer games and an organic light-emitting diode (OLED) to address such challenges. This approach was adopted to provide an enjoyable test experience, and results indicated that this solution is feasible for children.

### 3.4. Web and Smartphone-Based Tools

With the advent of modern technologies, several improvements have been made in visual function testing, leading to robust, time-efficient, reliable, and evidence-based screening tools [71]. Studies show the reliability of automated vision testing tools for objective measurements of eyes, including autorefractors, photo screeners, and smartphones-based devices [56]. Such tools can offer a broad range of vision tests, such as refractive errors, visual acuity, visual efficiency skills, color vision, stereovision, contrast vision, visual field test, and testing the oculomotor behavior of the eyes.

Today, health communities often use smartphone technologies in telemedicine for vision screening. Due to COVID-19, teleophthalmolog is being used by researchers for measuring visual acuity remotely [72]. Most of the available applications use optotypes for vision screening. "Peek Acuity" [73] uses the optotype letter "E" (different sizes, and rotated at different angles) for the visual acuity test. The Freiburg Vision Test (FrACT) [74] is a corresponding web-based application that also uses similar optotypes (often "E" or "C" of different sizes, and rotated at different angles) to evaluate visual acuity, and has significant sensitivity among preschool-aged children [75].

The use of "E" or "C" optotypes in web-based applications increases the objectivity of the test when the child cannot recognize the letter and there is no learning effect from one eye to the other. The web-based tests are easily accessible, validated, store data to electronic patient record, and can be shared remotely. Several computerized vision screening systems are portable and often provide a self-assessment environment without a need for the supervision of vision experts [76].

The main disadvantage of the tests is the presentation of one optotype without visual crowding effect (added distractors around a central target), which is very important in the assessment of visual acuity, the calculation of crowding ratio, and the diagnostics of amblyopia, dyslexia, learning disabilities, and ADHD [77,78]. Figure 3 shows the different types of technologies supporting vision screening.

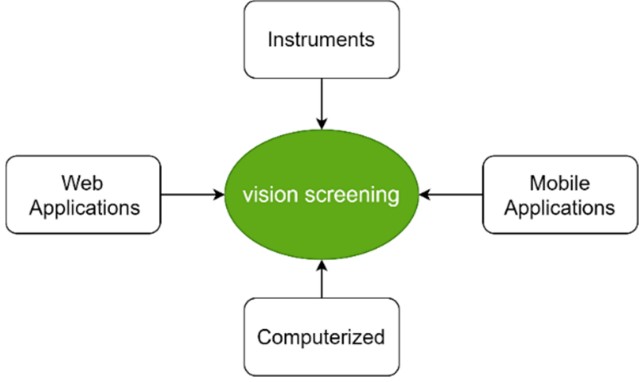

**Figure 3.** Different paradigms of technologies assisting experts in vision screening.

## 4. Visual Information Processing and Visual Cognition

Visual information processing is a cognitive skill to extract, organize, and analyze the visual information acquired from the environment [79]. Researchers believe that visual cognition is a complex process that involves visual information processing and cognitive factors. These work together to recognize the object from the scene using prior knowledge and retina input [80,81]. Scheiman [7] divides the information processing mechanism into three components: visual analysis, visual-spatial, and visual-motor processes. Figure 4 provides a basic description of these three components. These components represent specific skills of the individuals involved in performing tasks or activities. The literature argues that poor information processing skills such as visual-spatial and visual-motor integration, and visual analysis skills, are critical for school children [82].

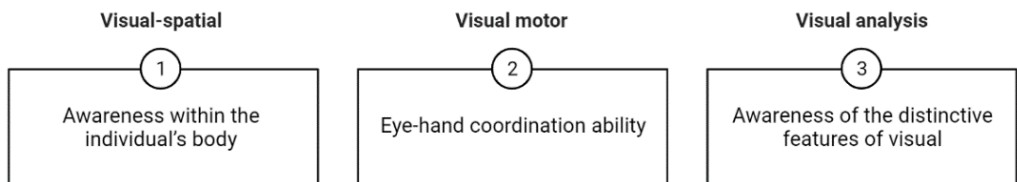

**Figure 4.** Visual information processing skills required for awareness and eye–hand coordination.

Vision experts test visual information processing skills in children using various objective or subjective standardized tests such as the Beery–Buktenica Developmental Test of Visual–Motor Integration (VMI), Motor-Free Visual Perceptual Tests (MVPT), and Rapid Automatized Naming (RAN) with the subset of the Developmental Eye Movement (DEM) test [82,83]. Studies have shown that children have visual processing disorders if they fail in such tests. Insufficient visual information processing is further associated with lower academic outcomes [79,84].

While eye-tracking technologies are helpful in recording eye movements such as fixations and saccades [85], which contribute to examining gaze movement in relation to visual representations, these relations are not necessarily enough to understand a whole picture of problems. For a better interpretation of the results, the relationships between cognitive skills are also important. Furthermore, handling the direct interpretations depends on prior knowledge [86] or usage of the context [87,88]. Other studies that may contribute to improved results would need a better understanding and interpretation of individual capabilities [89,90], hand–eye coordination [91], or children's adaptability [92] to different situations.

Information about the systems that observe brain activities [93,94], or advanced methodologies that create new technologies to support health [95], requires a high level of knowledge about information technologies and human cognition. According to what we understand, the use of eye-tracking technology is not widespread for diagnosing vision problems. However, it has great potential for complementing the work of vision experts by recording and analyzing the eye movements, and comparing the stored data of the subjects. The newest eye-tracking technology allows for controlling the distance from the screen, adjusting the position of the child's head to the test, and informing the assistant about a wrong position, which is very helpful for collecting the valid data.

## 5. Stakeholders Influencing Children's Vision Screening (RQ1)

This section identifies the main stakeholders in order to answer the RQ1: "Who are the stakeholders influencing vision screening at schools, and what are their roles?" Professional vision experts are not the only ones who perform the vision screening or help with managing vision problems. Other stakeholders that are involved or influence vision treatments include trained laypersons, school teachers, other educationalists, or parents [95]. In general, vision screening includes stakeholders from different domains who may have different routines and roles.

### 5.1. Vision Experts from Clinical Settings

Matta et al. [96] surveyed eighteen countries to examine the current state of international vision screening efforts worldwide. Their results show that 22% of vision screening is performed by orthoptists, 50% by ophthalmologists, optometrists, opticians, and parents, 17% by nurses, and 6% by doctors only. The primary stakeholders responsible for vision screening in children are qualified vision experts in clinical settings. The experts involved in children's vision screening, such as optometrists, ophthalmologists, and nurses, have the required expertise in clinical settings [97]. Their primary role is to diagnose vision problems in children that make them vulnerable during their development in general, and to vision impairments in particular [66].

### 5.2. Nurses, Parents, and Laypersons

A recent study by Sabri et al. [98] showed the result of non-healthcare professionals for vision screening. The accuracy of volunteers was 75% compared to that of optometrists, and they could be trained to perform vision screening on children. The literature review of Sharma et al. [14] identified many professionals and non-professionals such as nurses, teachers, health technicians, parents, and volunteers for vision screening for school-aged children.

In the United States, the results of school vision screening are often reported to parents or guardians, and they were instructed to follow up with an appropriate local eye-care professional [38]. Involvement from parents is crucial for identification and diagnosis, as well as for having a role in assisting the rehabilitation of children's functional vision.

### 5.3. Vision Teachers

Vision teachers (VT) are usually educated to Master of Science level in special pedagogy that focuses on children with visual impairment (low vision and blindness). A survey [99] found that the vision skills identified by VTs are mainly near and distance visual acuity, tracking, peripheral visual field, color perception, fixation, shift of gaze, contrast vision, and depth perception. The most common vision assessment method is a visual acuity test, which cannot detect a number of visual problems [39].

### 5.4. Other Specialists

Eye-care providers and primary care physicians have created several test batteries for vision screening in children, including cover tests, fixation, red-flex, acuity, color vision, and another test to identify oculomotor dysfunction [37,100,101]. As we have argued, several of these can be challenging to detect, in particular in relation to functional vision problems [100]. Recent studies indicate that 22% to 24% of school children in total, and 96% of children with a learning disability, have oculomotor dysfunction [102,103]. If the numbers are accurate, and the symptoms of oculomotor dysfunction and dyslexia can be the same, many children may receive the wrong diagnoses. Table 3 summarizes the stakeholders and their roles in children's vision screening.

**Table 3.** Papers grouped based on stakeholders and their roles.

| Studies | Stakeholders | Roles |
|---|---|---|
| [96,97] | Ophthalmologists, Optometrists, Orthoptists, Opticians, Nurses | Vision screening in clinical setting. |
| [14,44,104–107] | Educationalist, Vision Teachers | Vision screening in schools. |
| [14,96,98] | Volunteers, Parents | Non-health professionals supporting vision screening directly or indirectly. |

## 6. School-Aged Children's Vision Screening (RQ2)

To answer Research Question 2, "What is known about children's vision screening in schools?", we need to understand the cases where school children are screened and how studies report these results.

## 6.1. Vision Screening Programs for Schools

There are several countries with school vision screening programs, such as parts of the US, South Africa [39], and a county in Norway [18]. Across different countries, there is a substantial variation in screening programs focusing on vision and hearing. Studies show that only 21–36% of children younger than six years of age have received vision screening in the US [108].

A literature review [39] examining vision screening in 18 countries shows that many countries only perform partial vision screening; 88% are missing necessary tests, such as those indicating a need for eyeglasses (e.g., tests with Snellen or Tumbling "E" charts). In another study, Ciner et al. [109] highlighted the potential pitfalls in the screening process for preschool children, such as the agreement of effective methods of screening and the variability of test batteries in US states. Some studies reported the lack of professionals for vision screening in countries such as Turkey and Tanzania, and documented in reports from European countries [110–112]. Hopkins et al. [101] also argued that refractive error, strabismus, and amblyopia are most likely considered for screening in preschool and school entry children, whereas there is no agreed policy for the screening of binocular vision problems, functional vision, and other eye conditions in Australia and worldwide. While the work of non-professionals can be supportive and complimentary, their duties and roles are unclear at the moment. They supplement professionals in different ways and with different, often limited tests [39,42,112]. Sabri et al. [98] found that the accuracy of non-healthcare workers is 75% compared to professionals. However, non-professional volunteers can be trained to detect vision problems in children, especially if they have the right tools to support them.

## 6.2. Opinions on Vision Screening Programs at Schools

Screening for all schools may help identify children with vision problems as early as possible and increase awareness among school staff and parents about vision problems [113]. Among the benefits of school vision screening programs is improving the teacher–child relationship, and enabling teachers to know which support they need to provide [114]. This support also depends on how children can accept instructions, and their cognitive abilities. A basic vision screening can be performed in children's classrooms, which means that their school routine is not necessarily disturbed [111,115]. On the contrary, the training for school teachers may be perceived as an additional burden or a need to require additional resources [116]. For some schools, obtaining permission from the school authorities for holding an eye screening activity is a major challenge; school authorities often feel that the screening program interferes with their regular academic calendar and does not add value to the school or the children [117,118].

Teachers should collaborate with vision specialists, vision teachers, or occupational therapists to avoid misidentifying children with vision problems [110]. Regarding parental involvement, there should be announcements and discussions during the parent–teacher meeting in schools about the benefits and screening program structure prior to the screening [95]. After the screening, parents should be advised regarding the problems identified and the possibilities for intervention.

## 6.3. Opinions on Vision Screening for Many, and by Non-Professionals at Schools

Sudhan et al. [113] examined 530 schools with 77,778 children in India. While the teachers involved in the study may have complemented the screening of health professionals, the accuracy of their identification was not recognized [113]. Since the data collection mainly consisted of manual screening of various quality, it is not straightforward to relay or compare certain screening results from this study.

Collecting vision measurements would improve vision assessment protocols and support progress evaluation during rehabilitation, training, or visual therapy. Objective measurements are possible using ETs [33,119,120], but ETs alone are not enough [23]. We must consider new user-friendly, accessible, and cost-effective specialized software tools to

support the right interventions from stakeholders contributing to a broader investigation. However, as pointed out earlier, there is also a lack of stakeholders who can perform vision screening and assist rehabilitation.

Figure 5 depicts current recommendations suggested by the literature to strengthen the school vision screening programs. The purple box highlights the need for uniform screening methods [109,121], the red box shows that including volunteers would be more helpful in screening programs [98], the yellow box represents the recommendation to consider that functional vision screening should be included in screening programs [101]. The blue box recommends that school teachers could be trained to support vision screening [116]. Last, the green box shows all stakeholders involved in vision screening should communicate clearly and provide the information necessary to help vision screening [38,101].

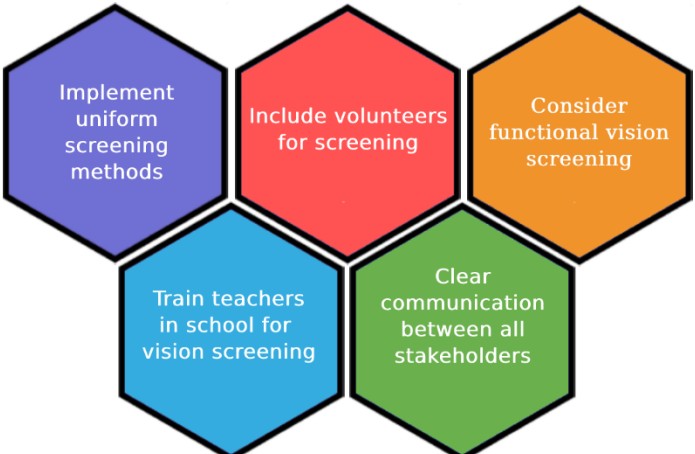

**Figure 5.** Current recommendations from the literature to fill the missing gaps in school vision screening programs. The colors in the figure depict recommendations for various methods and from various stakeholder groups.

## 7. ET Support for Vision Screening and Training (RQ3)

This section discusses Research Question 3: "How can ET better support screening and training vision?"

### 7.1. Usage and Types of ET Technologies

The use of eye-tracking technologies has significantly increased in cognitive neuroscience, the assessment of functional vision, applied engineering, education, and customer visual attention research [122–124]. In recent years, research studies have shown a potential use of eye-trackers to quantify visual information processing, visual function assessment (VFA), and diagnose nystagmus and strabismus vision problems [33,125].

The eye-tracking method aims to record the gaze direction at a specific time and understand visual attention. To achieve this aim, four types of eye-tracking techniques have developed over time. The classifications of four general ET techniques are: scleral search coil, electro-oculography (EOG), infrared oculography (IOG), and video oculography (VOG) [126]. Most modern eye-trackers use the video oculography (VOG) technique to record eye movements by applying the pupil center corneal reflection (PCCR) method [127]. The advanced image processing algorithms are also applied to the subject's eye position, relevant to stimuli [128]. The commercial eye-trackers are mobile, remote, and embedded in a head-mounted display (HMD), or add-ons for HMD. Remote eye-trackers require the participants to maintain head position throughout the entire process, while mobile or HMD eye-trackers do not have such conditions.

Generally, the refresh rate of eye-trackers varies from 30 Hz to 1200 Hz for recording eye movements, such as fixations and saccades. Eye trackers with low-frequency sampling have high latency in evaluating the point of gaze. The eye-trackers with a high-frequency sampling rate and low latency come with high-resolution cameras and they cost.

Several research studies showed the use of ET for vision screening, especially the visual functional assessment (VFA). Kooiker et al. [33] observed and quantified ET data to examine the oculomotor control and visual function assessment of visually impaired children. Murray et al. [35] demonstrated the reliability of ET by cluster analysis to determine the normative values of oculomotor metrics, such as saccades and fixations. Their results showed 85% of eye movement metrics were highly reliable and acceptable. Eide et al. [23] developed C&Look software that uses ET for complementing VTs and VT students to diagnose vision problems associated with oculomotor dysfunction in school children [111].

Furthermore, some research studies have shown the advantages of eye-tracking technologies for diagnosing strabismus and pathological nystagmus [129,130]. Table 4 shows the directions of ET used by researchers in the literature for vision screening.

**Table 4.** Overview of the studies that used ET for vision screening. N/R = not reported.

| Studies | Origin | Method | Vision Problem | Stakeholders | Subjects | Age |
|---------|--------|--------|----------------|--------------|----------|-----|
| [129] | Italy | Eye-tracker | Nystagmus | Ophthalmologists | 15 | N/A |
| [131] | India | Eye-tracker | OMD | N/R | 16 | 45–72 |
| [33] | Netherlands | Eye-tracker | OMD, visual functions | Orthoptists, Optometrists, Ophthalmologists | 126 | 1–14 |
| [130] | Thailand | Eye-tracker | Strabismus | N/R | 50 | 7–50 |
| [35] | U.S | Eye-tracker | OMD | Optometrists | 2993 | 5–62 |
| [132] | Eastern Europe | Eye-tracker | OMD | Ophthalmologists | 58 | 4–19 |
| [133] | South Africa | Eye-tracker | OMD | Clinicians | 33 | 21–24 |
| [79] | Australia | Eye-tracking, traditional | Refractive errors, binocular vision anomalies, OMD | Optometrist | 108 | 8–9 |

### 7.2. Recommendations for Challenges in ET Screening

While performing calibration or conducting the experiment with an eye-tracker, several factors can influence the accuracy and quality of the data being recorded, such as sitting position, body or head movements, and the person responsible for concluding the experiment [134]. Such factors are even more challenging for children and people with neurological disorders such as Alzheimer's disease, schizophrenia, stroke or autism [135]. Eye-tracking has been one of the main research methods in the last decades, not only for the diagnosis of eye movement disorders, but additionally for gaining insights into early (neuro)cognitive development. Through eye-tracking, gaze location can be objectively measured from children as young as a few days old, and up to adulthood.

Several research studies have recommended protocols and introduced new algorithms to mitigate the potential difficulties in the calibration or eye movements recordings process. Table 5 illustrates the recommendations extracted from the literature.

**Table 5.** Recommendations for calibration and screening for children and infants.

| Studies | Recommendations |
|---------|-----------------|
| [136] | Play cartoons to attract the attention of the child, use Rifton chair, caregiver support. |
| [137] | Apply linear transformation on fixation coordinates. |
| [134] | Use of animated calibration targets (looming or twisting), gaze coordinates should be accepted within first 4 s, keep changing background screen color of calibration scene. |
| [138] | Consider monocular calibration of each eye. |

### 7.3. Microsaccades

Microsaccades are small involuntary movements, including drift, tremor, and fixational saccades [139], and ranging from 0.5 to 2.5 Hz [140]. Studies reported that microsaccades help attain visual attention, processing cognitive tasks. In the non-visual cognition process, the frequency rate of microsaccade decreases and the magnitude of microsaccade increases with the task difficulty level [141]. Microsaccades can be detected with an eye-tracker, but with the cost of a high-frequency rate. Marcus et al. [140] analyzed the

data quality of microsaccade measurements collected by Tobii Pro Spectrum 1200 Hz and EyeLink 600 Hz eye-trackers. The results of their study showed that the high-frequency eye-tracker attained the microsaccades rate without missing any microsaccades.

### 7.4. Eye-Tracking Data Analysis

Eye-trackers offer high-frequency sampling rates. Therefore, investigating different parameters and conducting a statistical analysis of extensive data sets requires a set of tools. Many open-source and commercial software solutions are available (see Table 6) for generating heat maps, gaze points, blinking, fixation, microsaccade, and calculating the pupil size over time in an experiment.

**Table 6.** Names of open-source and commercial tools for recording and analyzing the experimental study using eye-trackers.

| Tool Name | Free | Paid |
|---|:---:|:---:|
| EventIDE [142] | | ✓ |
| Tobii Pro Lab [143] | | ✓ |
| SMI BeGaze [144] | | ✓ |
| iMotions [145] | | ✓ |
| Psychopy [143] | ✓ | |
| PyGaze [143] | ✓ | |
| OGAMA [146] | ✓ | |
| EMA Toolbox [147] | ✓ | |
| Gazepath [148] | ✓ | |
| PyTrack [149] | ✓ | |

## 8. Synthetizing the Results

### 8.1. Main Findings and Future Needs

The goal of this study is to gain an understanding of the current state of research in stakeholders involved in vision screening for children and the use of state-of-art eye-tracking technology to support vision experts, and to fill the gaps in the present literature on the use of eye-tracking technology for new possibilities and future lines of research. After answering the research questions, the following points synthesize the sometimes contradictory findings and serve as a basic departure for further research:

- For the first research question (i.e., who are the stakeholders influencing vision screening at schools?), seven primary vision experts and non-experts emerged from the literature: (i) orthoptists [33] (ii) optometrists [35] (iii) ophthalmologists [129] (iv) vision teachers [44] (v) nurses [150] (vi) parents (vii) volunteers [98].
- In most countries, routine vision screening starts at the age of 4 to 5 [49,151] or from 3 to 5 [95]. However, there are no uniform methodologies for vision screening [101,109,121].
- Many factors influence the low number of follow-up vision screenings for children, including parents' education, awareness, and social and economic factors [38,152,153]. Amblyopia (lazy eye) is often neglected in children because of the mentioned factors and the local medical system [154].
- Basic terms such as vision, screening, and technology mean different things in different fields. To find correct and accurate search terms is difficult and time-consuming.
- Vision problems affect children's social and academic performances, and can be a reason for students to drop out from educational institutes [19,28]. Therefore, it is crucial to assess children's vision throughout the school period at different stages [155].
- The examined documents reflect the current obstacles faced by children in their schools. At this age, the guardians of students, the teachers, and the child may not be aware of vision deficits. Consequently, concerns related to children's vision cannot be communicated well between the children and the guardians, and after that, if the children have possible vision problems, between the guardians and other involved stakeholders.

- An important observation from the literature and whitepapers is that vision care is often limited to the screening or handling of a specific vision problem. The support for treatment approaches and the responsibilities of stakeholders to aid visual deficits are lacking. Often, many stakeholders need to be involved in the treatment of vision problems, but their collaboration is fragmented [28,39].
- Stakeholders do not widely use eye-tracking technology, even though the literature has shown the promising applications and benefits of ET in diagnosing OMD, nystagmus, and strabismus [33,111,129,130].
- ET can collect eye movement metrics such as fixations, the number of fixations, saccades, microsaccades, and smooth pursuits [85,140]. The objective data collected from ET are reliable, and studies demonstrate that such metrics are helpful in functional vision assessment (FVA) [33,35].

### 8.2. Limitations

Several limitations have been identified in this study:

- We used Google Scholar and HVL Library as sources to investigate the literature. We may have missed some actual results that were not indexed in these two sources.
- The participants in the included studies were from different geographic locations, languages, and cultural backgrounds. The laws and regulations related to vision screening in schools and clinical may vary for each study, so comparing them is difficult.
- This was a rapid review, and thus was not systematic enough to identify more profound related domain (health education, and technology development and management knowledge) gaps for helping all school-aged children.
- This study used rapid review; therefore, results can be limited compared to systematic review where quantitative results can be provided.

### 8.3. Future Work

We used Google Scholar and HVL Library (ORIA) for the literature search. However, a systematic literature review is still needed to ensure the precise roles of stakeholders in screening and rehabilitation of children's vision. While this work has begun with the ambition to conduct a systematic literature review involving the domains promising better help for school-aged children with possible vision problems (e.g., education, health, and technology development and management), to perform such a review is difficult. The three domains, on their part focusing on vision disorders, are fragmented silos. Critical studies are complex to compare and contrast. This study identified several issues influencing the use of technologies supporting school-aged children's vision. To develop and use these technologies further systematic studies producing quantitative evidences on the benefits and limitations of the available technologies are needed (e.g., [17,33,37,156]).

Figure 6 shows the stakeholders scattered across different areas of research that are directly or indirectly involved in the general or functional vision screening and rehabilitation of children. A systematic literature review can provide evidence to support vision screening from such domains and stakeholders, and overcome the disconnect between such entities. However, such reviews must handle the different content, and compare and contrast. This will be difficult due to the many different sources and different types of evidence presented. Even the basic terms to use, such as vision, visual, screening, and technologies, have different meanings in the various associated fields. Vision (after relating the word to seeing or the eyes) can be much more than seeing or vision systems, especially in technology development and use, or in education. Screening automatically means examination by clinical experts in medical journals, but can additionally refer to data collection in engineering papers. Screening technologies can collect different types of information and data. Technologies can vary from tools such as using pens to examine vision to complex instruments or software programs helpful for clinical professionals.

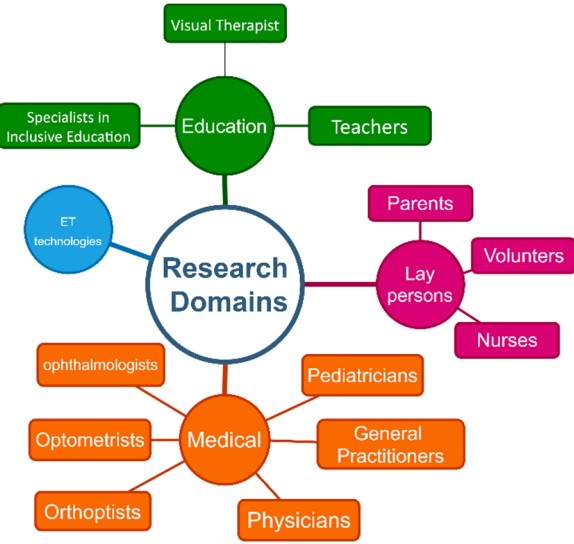

**Figure 6.** Research domains for further investigation.

## 9. Discussion

This paper investigated vision disorders in children, the stakeholders involved in vision screening for school-aged children, and focuses on identifying state-of-the-art eye-tracking technology for supporting vision screening. We identified important stakeholders contributions to children's vision health and discussed their role for collaboration. One of the main results is that many stakeholders with potential influences on managing children's vision are not systematically communicating with each other. Many non-specialist observers of possible vision problems, and the specialists diagnosing these problems, have difficulties initiating help for the children since they are not communicating. The potential for better communication needs to involve schools, where possible observers are, define the roles of responsibility, and involve other stakeholders, parents and professional vision experts to help children with vision impairment. A first step for better eye health, starting at schools, can be the recommendations summarized in Figure 5, according to several earlier literature (e.g., [6,15,111,115,117]).

Vision care is more than only diagnosing and screening children's vision. While research is pointing to training as an important part of treating vision disorders, training is often neglected and not connected to screening. This issue is reflected in the research on applying technologies. We exemplified the most common vision screening methods today, grouped into traditional, instrumental, and computerized tools, to better identify the potential of available tools and techniques in vision screening. However, we do not discuss a connection between screening and treatments, and if tools and instruments can be involved.

While the interest to use ET technology for diagnosing different vision impairments is growing, many of these technologies need to be improved to produce more understandable evidence regarding the different tests for non-specialists and for specialists. There are ET-based methodologies for analyzing eye movements; however, what exactly triggers the recorded eye movements, and what the relation is between attention and focus and the different parts of the vision system, should be further examined. The concrete measurements that ET can produce, including the fixation points, areas, number of fixations, saccades, smooth pursuits, and microsaccades from eye data, can be recorded and examined. Still, its role for visual information processing and the vision system needs better clarifications. This can only be achieved through the collaboration of vision experts and technology developers.

This study used Google Scholar and HVL Library (ORIA) [32] to find a broad range of published literature (peer-reviewed journals) and grey literature, including white papers and conference proceedings focusing on the areas of vision screening, education, technology,

and cognition. In this study, we used a rapid review approach following the eight phases defined by Khangura et al. [25] to collect and synthesize the pieces of evidence that provide a high level of understanding of current challenges in school-aged children vision screening programs. Defining research questions and performing the literature search was performed using those eight phases iteratively.

Across the world, many screening programs exist for preschool and school children. The literature demonstrates that there are still several gaps and limitations in the current screening procedure, including no adequate information on the ideal age of children for screening, inconsistent methods of screening in different states and regions, little agreement on effective methods of screening, insufficient resources and vision experts and an absence of functional vision assessments [101,109–112]. Intensive research is needed to see how to solve these problems, and the possibilities need to be investigated. A possibility that was investigated in this paper is empowering non-health-specialist volunteers and school teachers to perform vision screening where the intervention can be performed without a vision specialist, achieved by using technologies [14,98]. Professionals utilize traditional, instrumental, and computerized tools and technologies to conduct vision screening. Although traditional methods of screening are the gold standard, instrumental screening is gaining the attention of vision experts and is recommended by the American Academy of Ophthalmology (AAO), the American Academy of Pediatrics (AAP), the American Association of Certified Orthoptists (AACO), and the American Association for Pediatric Ophthalmology and Strabismus (AAPOS) for children under certain conditions [48,55]. Computerized technology, especially eye-tracker devices integrated with computer software, shows promising applications for collecting objective evidence of the eye movements that can assess children's functional vision [33,132]. While using ET, some factors such as calibration setup, the sitting position of children, and the adjustment for children with special needs can influence data quality. Still, research studies have demonstrated that adopting various techniques and algorithms could optimize data quality and minimize the mentioned challenges [136].

## 10. Conclusions

This paper revealed how stakeholders, improved communication, and technologies can better support vision screening for school-aged children. One of the most important current challenges is to involve schools in these activities. This involvement requires uniform screening methods that include functional vision screening and possibilities to follow up vision problems. Current technologies, including eye-tracking technologies, have great potential for supporting these methods, providing measurable results, and bridging over the fragmented communication between the involved stakeholders.

**Author Contributions:** Q.A. was the primary author responsible for investigation, data curation, visualization, and writing—original draft preparation. Supervision, conceptualization, methodology, writing—review and editing, I.H.; supervision, writing—review and editing, and methodology, C.G.H.; validation, writing—review and editing, G.K.; validation from vision science and optometry, C.C.; validation from cognitive science, writing—review and editing, A.K.; writing—review and editing, J.K.; writing—review and editing, S.T. All authors have read and agreed to the published version of the manuscript.

**Funding:** The project is sponsored by SECED, funded by the Research Council of Norway, project no: 267524/H30. The project is also sponsored by EFOP-3.6.1-16-2016-00003 funds, consolidatation of long-term R and D and I processes at the University of Dunaujvaros.

**Institutional Review Board Statement:** Not applicable.

**Informed Consent Statement:** Not applicable.

**Conflicts of Interest:** The authors declare no conflict of interest.

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
