# Peer review of "Current Challenges Supporting School-Aged Children with Vision Problems: A Rapid Review"

_applsci, doi:10.3390/app11209673_

Round 1

Reviewer 1 Report

Authors made a mapping study about the current challenges supporting vision screening of school aged children.

This is a very interesting topic, but, in my opinion, authors need to clarify some aspects to go ahead with the publication.

First of all there are some aspects regarding the format:

  1. Type of article: please confirm if it is an article or a review
  2. Abstract: It does not follow the journal guidelines
  3. If the authors confirmed that the type of article is "article" they should reconsider the sections

Concerning the content of the manuscript:

  • Introduction.

It would be interesting to introduce the different types of reviews and include in the aim of the study if it is a mapping study or not (it is not clear)

  • Methodology:

I think that figure 1 should included the number of paper examined in each step. 

Authors should clarify the keyword used in the search. Are they the ones describes in line 124 or those in line 164?

  • Sections: section 5 and 6 has the same name. It is confused, please change it
  • Discussion

It should start with the strengths of the research

It is important to discuss the different types of reviews, justifying why they have chosen a mapping study.

It would be also important to discuss why the use google scholar instead of other databases.

This section should finished with a conclusion supported by the findings of the study

Reviewer 2 Report

I consider this manuscript to have valuable information that would be of interest to the journal's readers. However, it needs a major revison, specific:

1. Introduction section; the paragraph from "the examination of the peer-reviewd... to ... can ease the expert's work" should be moved to the discussion section. Next paragraph from "the paper is structured... to ... Section 8" should be removed. 

2. Please remove Tables 1,2,3 and add new Table with the data about screening for OMD in different countries (Part 5) or listing the studies from different countries with subjects' age 

3. Instead of Table 4 draw new Figure. (finally 6 figures and 1 table)

4. There is no need to divide RQ2 in two separate parts 5 and 6 

 5. The discussion should to be re-written - please add paragraph from the introduction (remark 1) and focus on the comparison between studies.

6. You wrote the limitations but you forgot about conclusions.

Reviewer 3 Report

The authors state that the aim of the paper is ... "to contribute to a better understanding of the importance of vision screening of school aged children and investigate the possibilities of eye tracking technologies". That eye examinations of school-aged children are important is undisputed and not new knowledge. It is also no news that many technical devices are available for this purpose.  From a paper about ophthalmologic examinations in school children one should expect that the kind of visual problems and eye movement disorders that are frequent in school age children are described, which causes and risk factors exist, how well children tolerate which measuring methods and which sources of error exist especially in learning disabled and restless children. However, the paper does not go much beyond the demand for ophthalmological investigations and the statement that there are several measurement methods.

Lines 377 ff: The authors mention several eye tracking technologies, but do not discuss issues of application in children. For example, there is the question of how well children tolerate the measuring devices and whether and to what extent abrupt head and body movements in restless children interfere with the measurement. Often the devices slip so that the calibration of the measurement is not granted. Eye movement disorders are by no means all caused by disorders of the eye muscles. Eye movement disorders occur after dysfunctions in the area of the brain stem where the eye muscle nuclei are located, after dysfunctions in the area of the cerebellum, frontal regions and the parietal and occipital area of the cerebral hemispheres. In reading, it is still debated whether eye movements deviating from the norm are the cause of poor reading or the consequence of other disorders that reduce reading ability. All these problems of the practical application of devices for the examination of eye movement disorders as well as questions of the connection of eye movement disorders with school achievement impairments are not addressed in the paper.

Lines 342ff, 344 ff, 353 ff: Also in other investigations of visual performance like the measurement of visual acuity it is only stated that there are modern methods of measurement but their advantages and disadvantages are not discussed. Example: In the determination of visual acuity only one optotype is shown in the fixation point. However, reading does not mean letter by letter reading by moving one letter after another in the fixation point. Instead, a sequence of letters must be read at a time. This requires sufficient visual acuity to an eccentricity of about 5 degrees on the horizontal meridian. Such requirements, for which the measurement of acuity on the fixation point is not sufficient, are discussed in the literature but are not mentioned in the paper.

Lines 328 ff, Another example is the assessment of the visual field. Here it should be discussed in which children a visual field measurement is useful. In children who have not had retinal disease and in whom no brain damage is suspected,visual  perimetry is not indicated. However, many children under the age of 10 years in whom the visual field including luminance difference thresholds should be assessed, do not fixate steadily, so that automatic perimetry leads to significant errors. In this case, hand perimetry (Goldmann or equivalent) or, if this is also not possible, arch perimetry can be performed. However, the latter cannot determine visual field defects and luminance difference thresholds in areas close to the fixation point.

In summary, the paper does not meet the requirements of a scientific paper. Therefore, a publication cannot be recommended.

Round 2

Reviewer 1 Report

I would like to thanks the authors for such a deep review of the article.

I thinks that it has improve a lot. It is easy to read and clear, with a very interesting sythesis of the results.

Thanks for considering all the suggestions

Author Response

Thank you for your valuable review of the paper.

Reviewer 2 Report

I am partly satisfied with authors reply. I would suggest minor revision. Specific: please join together the current discussion and conslusion sections and then write the conclusion section again with maximum three sentnces of simple summary. 

Author Response

Thank you for your comments and suggestions.

We did the following changes.

1. Page 18, Lines (739-746), We wrote a conclusion paragraph as you suggested.

Reviewer 3 Report

The paper can be recommended for publication in the present version.

Author Response

(The authors gave the same response as above.)
